# Alkaline ceramidase catalyzes the hydrolysis of ceramides via a catalytic mechanism shared by Zn$^{2+}$-dependent amidases

Jae Kyo Yi[1,2¤], Ruijuan Xu[1,3], Lina M. Obeid[1,3], Yusuf A. Hannun[1,3], Michael V. Airola[2☯]*, Cungui Mao[1,3☯]*

1 Department of Medicine, Stony Brook University, Stony Brook, NY, United States of America,
2 Department of Biochemistry and Cell Biology, Stony Brook University, Stony Brook, NY, United States of America, 3 Stony Brook Cancer Center, Stony Brook, NY, United States of America

☯ These authors contributed equally to this work.
¤ Current address: Department of Radiation Oncology, Dana-Farber Cancer Institute, Boston, MA, United States of America
* Cungui.Mao@stonybrookmedicine.edu (CM); Michael.Airola@stonybrook.edu (MVA)

**Data Availability Statement:** All relevant data are within the paper and its Supporting Information files.

## Abstract

Human alkaline ceramidase 3 (ACER3) is one of three alkaline ceramidases (ACERs) that catalyze the conversion of ceramide to sphingosine. ACERs are members of the CREST superfamily of integral-membrane hydrolases. All CREST members conserve a set of three Histidine, one Aspartate, and one Serine residue. Although the structure of ACER3 was recently reported, catalytic roles for these residues have not been biochemically tested. Here, we use ACER3 as a prototype enzyme to gain insight into this unique class of enzymes. Recombinant ACER3 was expressed in yeast mutant cells that lack endogenous ceramidase activity, and microsomes were used for biochemical characterization. Six-point mutants of the conserved CREST motif were developed that form a Zn-binding active site based on a recent crystal structure of human ACER3. Five point mutants completely lost their activity, with the exception of S77A, which showed a 600-fold decrease compared with the wild-type enzyme. The activity of S77C mutant was pH sensitive, with neutral pH partially recovering ACER3 activity. This suggested a role for S77 in stabilizing the oxyanion of the transition state. Together, these data indicate that ACER3 is a Zn$^{2+}$-dependent amidase that catalyzes hydrolysis of ceramides via a similar mechanism to other soluble Zn-based amidases. Consistent with this notion, ACER3 was specifically inhibited by trichostatin A, a strong zinc chelator.

## Introduction

Ceramidases catalyze the hydrolysis of ceramides, which are important intermediates of complex sphingolipids that play an important role in the integrity and function of cell membranes [1], resulting in the formation of free fatty acids (FAs) and sphingosine (SPH). Alkaline ceramidase 3 (ACER3) is one of three ACERs (ACER1, 2, and 3), which were cloned by our group

**Funding:** This study was funded by the Foundation for the National Institutes of Health under the following grant numbers: R01CA163825, R01GM130878-02, and R01GM130878-03 to CM, R01GM130878-01 to LMO, P01CA097132 to YAH, and R35GM128666 to MVA.

**Competing interests:** The authors have declared that no competing interests exist.

[2]. ACER3 is localized to both the Golgi complex and endoplasmic reticulum (ER), and it is highly expressed in various tissues compared with the other two alkaline ceramidases [3]. Additionally, it was demonstrated that ACER3 hydrolyzes a synthetic ceramide analogue NBD-C$_{12}$-phytoceramide *in vitro*; thus, it was formerly named the human alkaline phytoceramidase (haPHC) [2]. Later, we reported that ACER3 catalyzes the hydrolysis of ceramides, dihydroceramides, and phytoceramides carrying an unsaturated long acyl chain (C18:1 or C20:1) with similar efficiency [4]. Recently, important roles of ACER3 in neurobiology have been highlighted. We demonstrated that ACER3 deficiency results in Purkinje cell degeneration in mice [5] and that a point mutation of ACER3 leads to progressive leukodystrophy in early childhood in humans [6].

The sequence similarity between ACERs, and progestin and adipoQ receptors (PAQR receptors) was revealed by Villa *et al.* [7]. In turn, ACER3 has been reported to belong to a large protein superfamily referred to as CREST (for alkaline ceramidase, PAQR receptor, Per1, SID-1, and TMEM8) [8]. The CREST superfamily is highlighted in various cellular functions and biochemical activities. Ceramidases are lipid-modifying enzymes [9]. The PAQR receptors contain the ADPRs, which regulate energy metabolism and have also been reported to hydrolyze ceramide [10–13]. The Per1 family consists of fatty acid remodeling hydrolases for GPI-anchored proteins [14, 15]. SID-1 family of putative RNA transporters is involved in systematic RNA interference [16, 17]. Lastly, the TMEM8 family is known to play a role in cancer biology as putative tumor suppressors [18, 19].

All members of the CREST superfamily conserve a set of three Histidine residues, a Serine residue and an Aspartate residue. These residues are the defining motif of the CREST family and were suggested to form a metal-dependent active site for lipid hydrolysis. However, the specific roles of the conserved residues in CREST proteins have not been experimentally proved. The independent discoveries of hydrolase activities in alkaline ceramidases [2, 3] and Per1 [15] suggest that the majority of CREST members are metal-dependent hydrolases. In support of this, the ADPRs were recently found also to hydrolyze ceramide, but at a slow rate [13].

The crystal structure of ACER3 has been recently reported [20]. It was confirmed that ACER3 has the seven transmembrane domain architecture and a catalytic Zn²⁺ binding site in its core. Interestingly, the Ca²⁺ binding site was found to be physically and functionally connected to the Zn²⁺, implying that Ca²⁺ regulates ACER3 enzymatic activity. Molecular dynamic (MD) simulation was also used to evaluate the molecular mechanism of ACER3 enzymatic function.

Compared with the biological role of ACER3, which has been studied in the last decade, the kinetic mechanism of its intrinsic ceramidase activity has not been described in detail. In this study, our biochemical results and mutational analysis allow us to propose the mechanism for how ACER3 hydrolyzes ceramides, which most likely applies to other CREST members. Furthermore, the inhibitor assay exhibits a possibility to develop direct and specific inhibitors of ACER3 and its related paralogs, ACER1 and ACER2, for treatments of diseases associated with dysregulation of the metabolism of ceramides and other sphingolipids, such as cancers, diabetes mellitus, cardiovascular diseases, neurodegenerative diseases, and skin diseases [5, 6, 21, 22].

## Materials and methods

### Site-directed mutagenesis

The yeast expression plasmid pYES2-ACER3 containing the FLAG-tagged open reading frame (ORF) of the wild-type *ACER3* gene was constructed in our previous study [2, 3] and used as a

template for site-directed mutagenesis. The codon of His81, His217, His22, Asp92, Ser77 in the ACER3 ORF in pYES2-ACER3 (WT) was switched to a codon of Ala or Cys using Quick-Change II XL, a Site-Directed Mutagenesis Kit (Agilent Technology; Danbury, CT) following the manufacturer's manual. The resulting plasmids (H81A, H217A, H221A, D92A, S77A, and S77C) containing the mutated ORF were sequenced to confirm the codon switch.

## Protein expression in yeast cells

The *Saccharomyces cerevisiae* mutant strain Δ*ypc1*Δ*ydc1*, in which both yeast alkaline ceramidase genes *YPC1* and *YDC1* was deleted [2], was grown at 30˚C on agar plates containing YPD medium (1% yeast extract, 2% peptone, 2% dextrose), or in liquid YPD medium with rotational shaking at 200 rpm. Δ*ypc1*Δ*ydc1* cells were transformed with WT, H81A, H217A, H221A, D92A, S77A, and S77C, respectively, and the resulting transformants were selected on agar plates containing uracil-dropout synthetic medium (SC-Ura) with 2% glucose. Δ*ypc1*Δ*ydc1* yeast cells transformed were grown in SC-Ura medium with 2% galactose to induce the expression of ACER3 or mutant ACER3s, respectively. Total membranes were prepared from yeast cells, and the expression of wild-type or mutant ACER3s was determined by Western blot analyses using anti-FLAG antibody as described [3].

## Sample preparation and enzyme reaction for the ceramidase activity assays

The total membranes were collected from transformed yeast cells as described in [3]. Approximately 10 mL of cultured cells were pelleted at 3000 rpm 4˚C for 5 min, washed twice with 1.5 mL of PBS. The washed cells were resuspended and swollen in 1.0 mL of lysis buffer (25 mM Tris/HCl, pH 7.4, and 5 mM CaCl$_2$) containing 1 mM PMSF and 1× complete protease inhibitor cocktail (Roche, Indianapolis, IN). Then, cells were homogenized with 35 strokes in a tight-fitted Dounce homogenizer. Crude homogenates were centrifuged at 1000 x g at 4˚C for 5 min to remove the nucleus and unbroken cells. The supernatant was further centrifuged at 20,000 x g at 4˚C for 30 min to remove heavy organelle fractions. Cell membranes were pelleted by ultracentrifugation (at 100,000 g for 1 hr with a TLA55 rotor, Beckman Coulter Life Sciences, Brea, CA) and resuspended in membrane homogenate buffer (25 mM Tris, pH7.4, 5 mM CaCl2, and 150 mM NaCl) by brief sonication for further analysis.

Membrane homogenates (0.5 to 2 μg of protein per reaction) were measured for alkaline ceramidase activity using several concentrations (2.5 to 400 μM) of D-$_{ribo}$-C$_{12}$-NBD-phytoceramide (NBD-C$_{12}$-PHC, Avanti, Alabaster, AL) as a substrate. Briefly, NBD-C$_{12}$-PHC was dispersed by water bath sonication in Buffer C (25 mM glycine-NaOH, pH 9.4, 5 mM CaCl$_2$, 150 mM NaCl, and 0.3% Triton X-100). The lipid-detergent mixtures were boiled for 30 s and chilled on ice immediately to form homogeneous lipid-detergent micelles, which were mixed with an equal volume of membranes suspended in Buffer B. Enzymatic reaction was carried out at 37˚C for 30 min. The extraction solvent (chloroform: Methanol, 1:1) was added with 3 volumes of reaction mixture to quench the reactions. After centrifugation, the organic phase was collected and dried under a stream of inert nitrogen gas. Dried samples were dissolved in Mobile Phase B consisting of 1 mM ammonium formate in methanol containing 0.2% formic acid for high-performance liquid chromatography (HPLC).

## Thin layer chromatography (TLC)

A thin layer chromatography (TLC) method was used to measure ACER3 ceramidase activity as the previous study [2] described. Briefly, 10 μl of each reaction mixture was spotted onto a TLC plate, developed in a solvent system consisting of chloroform, methanol, and 25% ammonium hydroxide (90:30:0.5). The TLC plate was dried and scanned by an imaging system

(Typhoon FLA 7000, GE Healthcare Life Sciences, Pittsburgh, PA) set in the fluorescence mode. The fluorescent band of NBD-C$_{12}$-fatty acid (NBD-C$_{12}$-FA) released from NBD-C$_{12}$-PHC was identified according to the standard NBD-C$_{12}$-FA spotted on the same TLC plate.

## High-performance liquid chromatography (HPLC)

HPLC was utilized to measure ceramidase activities and kinetics of ACER3 with a modification of the technique described in [23]. Ten μL of each reaction was injected into reverse phase liquid chromatography and separated by a Spectra C8SR Column (150 × 3.0 mm; 3 μm particle size; Peeke Scientific, Redwood City, CA). Mobile Phase A consisted of 2 mM ammonium formate and 0.2% formic acid in water. Mobile Phase B consisted of 1 mM ammonium formate and 0.2% formic acid in methanol. Fluorescence was determined using an 1100 Series HPLC-FLD Fluorescent Detector (Agilent, Santa Clara, CA), with excitation and emission wavelengths set at 467 and 540 nm, respectively. Fluorescent NBD-C$_{12}$-FA and NBD-C$_{12}$-PHC peaks of samples were identified by comparing their retention times with those of NBD-C$_{12}$-FA and NBD-C$_{12}$-PHC standards. Calibration curves were calculated by linear regression. The data were exported to Excel spreadsheets to generate calibration lines and calculate sample concentrations.

## ACER3 inhibitor assay

Before enzymatic reaction was performed, collected microsomes were pre-treated with inhibitors including DMSO (control), Trichostatin A (TSA, Sigma-Aldrich, St. Louis, MO), or C$_6$-Cer-Urea (Avanti, Alabaster, AL) for 30 min. The lipid-detergent mixtures were then mixed with an equal volume of microsomes. After incubation at 37°C for 30 minutes, the reaction was terminated by adding the extraction buffer (chloroform/methanol, 1:1). Dried samples were dissolved in Mobile Phase B for HPLC.

## Determination of kinetic constants

Kinetic constants were calculated from the Michaelis-Menten representations of at least three ACER3 activity assays in the absence of any inhibitor or in the presence of different concentrations (30, 60, and 90 μM) of TSA. The inhibition type of TSA was predicted using a Lineweaver–Burk plot.

## Protein expression analysis

Protein expression was assessed by Western blotting analyses using an anti-FLAG antibody (Sigma-Aldrich, St. Louis, MO) and anti-rabbit IgG (Cell Signaling Technology, Inc., Danvers, MA) secondary antibody.

## Results

### Comparison to ADPR structure

To investigate the catalytic mechanism of the CREST superfamily, we sought to combine the structural information available for the ACER3 and ADPRs, with the biochemical assays available for alkaline ceramidases. We chose to focus our study on ACER3 due to its robust ability to hydrolyze the fluorescent lipid substrate NBD-C$_{12}$-PHC, which can be easily quantitated.

The crystal structure of ACER3 and ADPR2 was adopted from the previous studies [20, 24] (Fig 1A and 1B). Both human ACER3 and ADPRs contain 7-transmembrane architecture with a canonical Zn-dependent hydrolase active site located at the bottom of a central cavity [20, 24] (Fig 1A and 1B). Multiple sequence alignment was performed with the amino acid

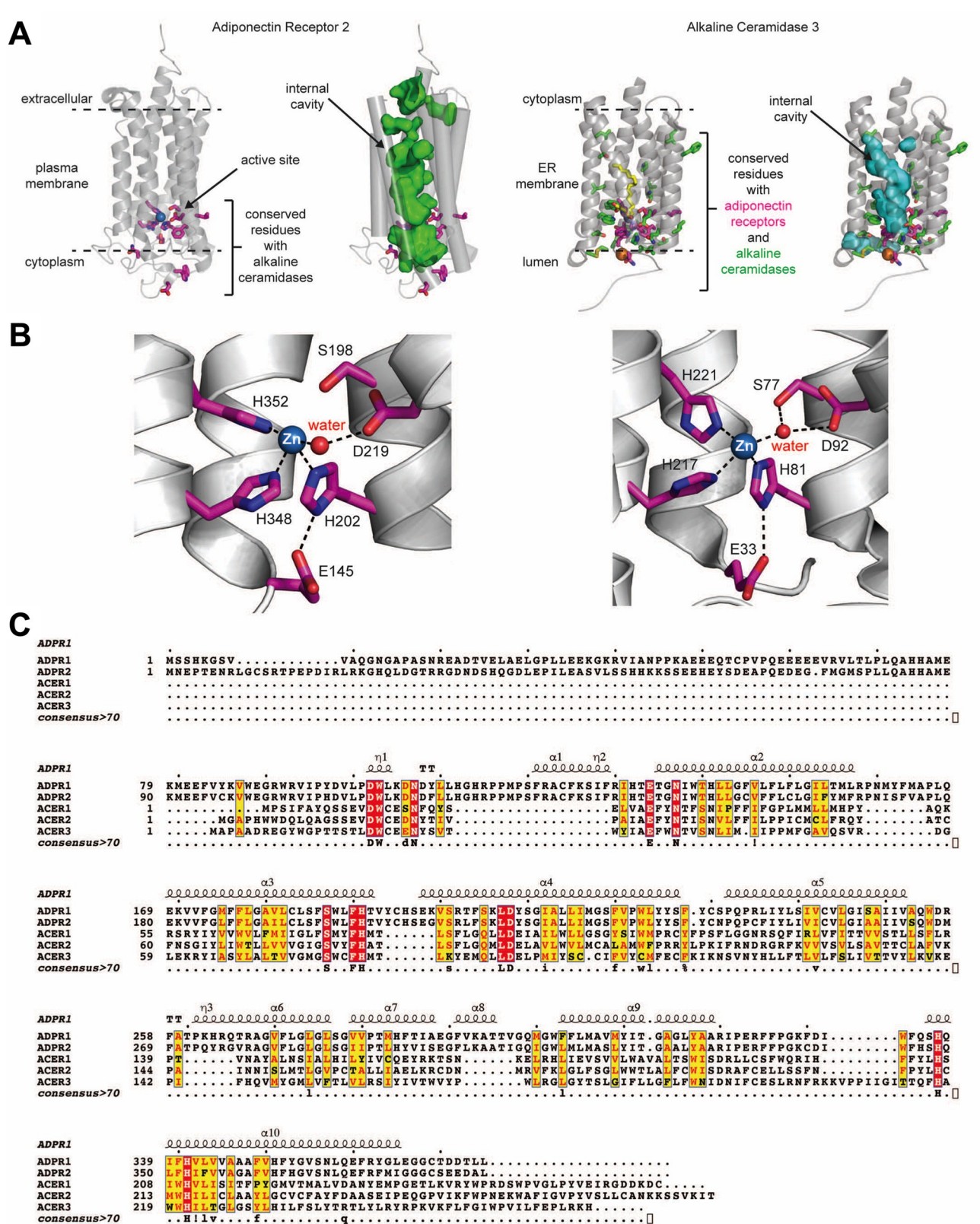

**Fig 1. Structure of ADPR2 and ACER3.** (A) The structure of ADPR2 and ACER3 [20, 24]. Conserved residues between ACER3 and ADPR2 in the active site are colored magenta. Conserved residues in ACER3 with other ACERs (ACER1 and ACER2) are indicated as green. (B) Active site of ADPR2

(left) or ACER3 (right) highlighting the Zn-coordinating side chains. (D) Multiple amino acid sequence alignment of ACER1, ACER2, ACER3, ADPR1 and ADPR2. The amino acids conserved in all proteins are boxed by red.

sequences of ACER1, ACER2, ACER3, ADPR1, and ADPR2 for comparison (Fig 1C), demonstrating the strictly conserved amino acids around the catalytic site. The ACER3 consists of a 7-transmembrane architecture with a central hydrophobic cavity similar to ADPRs. The twelve amino acids conserved with ADPRs are all clustered on the luminal side of ACER3, either present in the hydrophobic membrane-spanning segment or solvent exposed to the lumen. This includes the critical catalytic residues consisting of the three His residues involved in Zn-coordination, the Asp residue that bonds to the Zn-bound water molecule via a hydrogen bond, and the Ser residue that is universally conserved in the CREST superfamily (Fig 1C). The remaining conserved residues are either clustered around the core catalytic residues and involved in transmembrane helix-helix stabilizing interactions or are exposed to solvent in the lumen.

We also analyzed the position of the residues absolutely conserved among alkaline ceramidases but not conserved with ADPRs (Fig 1B and 1C). The majority of these residues clustered around the key catalytic residues within the active site, suggesting they may play a role in binding and recognizing the ceramide substrate. Interestingly, only a few residues are conserved within the upper portion of the hydrophobic cavity. The general lack of conservation in this region may be due to the specific preference of each ACER for different fatty-acyl chain lengths of ceramides, which are predicted to bind within this hydrophobic cavity.

## Kinetic characterization of ACER3 and its mutants

To investigate the kinetics of ACER3 ceramidase activity, we used a previously established microsomal assay in conjunction with a newly established HPLC assay with improved sensitivity. First, microsomes were purified from *S. cerevisiae* cells (*ΔYdc1ΔYpc1*), overexpressing human ACER3. Since *S. cerevisiae* has only two ceramidases, Ypc1p and Ydc1p, *ΔYpc1ΔYdc1* cells lack all endogenous ceramidase activity and thus provide the ideal signal to noise ratio to analyze ceramidase activity [2, 25]. In accordance with previous studies [2, 3], the ceramidase activity of ACER3 toward NBD-C$_{12}$-PHC was determined at pH 9.4, unless otherwise mentioned. Prior to determining the enzyme kinetics of ACER3, the linear detection limit of the product NBD-fatty acid was determined (S1 Fig). This newly established method was extremely sensitive and could quantitate NBD-FA levels at far lower levels than conventional TLC-based method (S2 Fig). In addition, the time and protein concentration for the reaction were optimized and confirmed to be in the linear range (S3 and S4 Figs). A plot of the reaction velocity (V) as a function of the substrate concentration (S) obeyed Michaelis-Menten kinetics ($R^2$ = 0.98, Fig 2A). The K$_M$ and V$_{MAX}$ values for NBD-C$_{12}$-PHC were calculated to be 15.48 ± 1.248 μM and 46.94 ± 0.8976 pmol/min/mg, respectively (Fig 2A).

The active site of ACER3 includes three Zn-coordinating His residues and an Asp residue that was hydrogen-bonding with Zn-bound water. We investigated the importance of each active residue of ACER3 by site-directed mutagenesis. Each residue was replaced with Ala to make the point mutants of ACER3, including H81A, H217A, H221A, and D92A. The expression levels of wild-type and all mutant ACER3 were verified to be similar by western blot (Fig 2B). Ceramidase activities of wild-type and the four-point mutants toward NBD-C$_{12}$-PHC were measured by the HPLC method (Fig 2C). Ceramidase activity was detected in none of the mutants. This confirmed again that these residues were all essential for ACER3's activity of hydrolyzing NBD-C$_{12}$-PHC. This result was verified by TLC, which showed consistency with the HPLC result (Fig 2D and 2E).

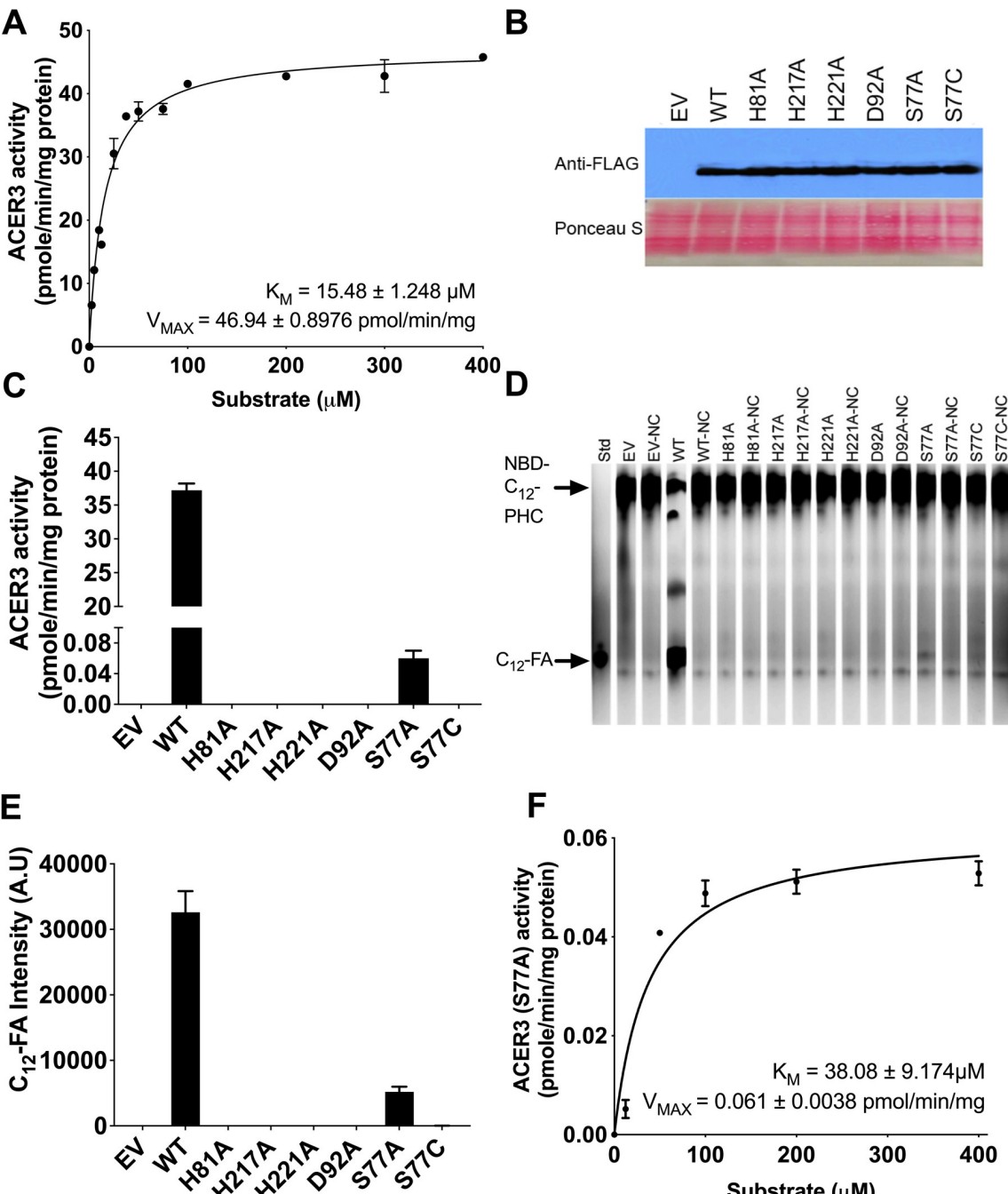

**Fig 2. The kinetics of ACER3 and mutants.** (A) Microsomes from yeast cells (*ΔYpc1ΔYdc1*) overexpressing wild-type ACER3 were measured for in vitro ceramidase activity on various concentrations of NBD-C12-PHC. $K_M$ and $V_{MAX}$ values are indicated. Reactions were conducted with 1 μg of microsome at 37°C for 30 min. Data represent the mean ± S.D. of three independent experiments performed in duplicate. (B) Western blot analysis verifying the expression levels of each mutant and wild-type ACER3. For normalization of equal loading, the membrane blot was stained by ponceau S. (C) Microsomes from wild-type and each mutant ACER3 were subjected to alkaline ceramidase activity using NBD-$C_{12}$-PHC. The release of the fluorescent product NBD-$C_{12}$-FA from the substrate NBD-$C_{12}$-PHC was detected by HPLC. (D) Microsomes from wild-type and each mutant ACER3 were subjected to alkaline ceramidase activity using NBD-$C_{12}$-PHC. The release of the fluorescent product NBD-$C_{12}$-FA from the substrate NBD-C12-PHC was detected by TLC. NC, Negative Control (heated microsomes) (E) Intensity of $C_{12}$-FA in (D) was measured by ImageJ (NIH, 1.53a) (F) Microsomes from yeast cells (*ΔYpc1ΔYdc1*) overexpressing mutant ACER3 (S77A) were measured for in vitro ceramidase activity on various concentrations of NBD-C12-PHC. $K_M$ and $V_{MAX}$ values are indicated. Reactions were conducted with 1 μg of microsome at 37°C for 30 min. Data represent the mean ± S.D. of three independent experiments performed in duplicate.

## Mutational analysis of Ser77

Unlike other Zn-dependent amidases, ACER3 contains a unique serine residue near the Zn$^{2+}$ ion in the active site. This Ser residue is universally conserved in the CREST superfamily and is part of the motif that defines this protein superfamily. However, the role of S77 is not clear as other Zn-dependent amidases do not contain a Ser residue in their active site. To ascertain the role of Ser77, we used site-direct mutagenesis to replace Ser77 with Ala (S77A) and determined the effect on ACER3 ceramidase activity. Interestingly, S77A had a low but measurable ceramidase activity of approximately 0.17% of that of WT ACER3 (Fig 2C–2E). The reaction of S77A followed Michaelis–Menten kinetics with a dramatically reduced V$_{max}$ (0.061 vs. 47 pmol/min/mg) and a slight increase in K$_M$ from 15.5 μM to 38 μM (Fig 2F).

To further investigate the role of Ser77, we generated the point mutant S77C using site-directed mutagenesis. This point mutant was chosen because we reasoned that the thiol group (-SH) of cysteine might complement the hydroxyl group (-OH) of serine. However, the S77C ACER3 point mutant had no measurable alkaline ceramidase activity (Fig 2C–2E). Since the predicted pKa of the cysteine thiol group is 8.14 and we conducted our analysis at the ACER3 pH optima of 9.4, we evaluated if lowering the pH to 7.5 would protonate the thiol group and eliminate the negative charge, would be sufficient to restore some ACER3 activity. Interestingly, decreasing the pH to 7.5 showed an increase of S77C activity, equivalent to the S77A mutant (Fig 3A). At pH 7.5, wild type ACER3 had a slight 7% reduction of activity and there were no measurable effects on all other mutants (Fig 3A). The K$_M$ and V$_{MAX}$ values of the S77C mutant were determined to be 66.43 ± 6.143 μM and 0.0337 ± 0.01 pmol/min/mg, respectively, showing that the plot obeys Michaelis-Menten kinetics (Fig 3B). This result was further verified by TLC, which showed partial ACER3 activity of S77C (Fig 3C and 3D). These results suggest that the thiol group of cysteines partially complement the hydroxyl group (-OH) of serine.

## Pharmacological inhibition of ACER3

There has been an effort to seek specific inhibitors of alkaline ceramidases. Additionally, HDAC (class I/II) inhibitors are well known for inhibiting Zn$^{2+}$-dependent amidases [26]. We decided to test if these inhibitors have an effect on ACER3 ceramidase activity. Microsomes from *S. cerevisiae* cells overexpressing wild-type ACER3 were treated with inhibitors including C$_6$-urea-ceramide (a neutral ceramidase inhibitor [27]), DMAPP (an alkaline ceramidase inhibitor [28]), and TSA (an HDAC inhibitor [29]). Neither C$_6$-urea-ceramide nor DMAPP affected ACER3 activity at concentrations up to 30 μM (S5 Fig). In contrast, TSA inhibited the activity in a concentration-dependent manner (Fig 4A) and the IC50 of TSA was estimated to be 71.41 μM. To determine the mechanism of inhibition (MOI) of TSA, we performed kinetic assays (Fig 4B). As seen in Fig 4C and 4D, TSA treatment raised the K$_M$ value while lowering the V$_{MAX}$ value, suggesting that TSA acts as a mixed inhibitor of ACER3 by binding free and substrate-bound ACER3 with different affinities.

## Discussion

Based on our mutational analysis, we propose a general acid-base catalysis mechanism for ceramide hydrolysis by ACER3. By our suggested mechanism, Asp92 is predicted to activate the water molecule by deprotonation, serving as a general base (Fig 5). The activated water molecule undergoes a nucleophilic attack on the ceramide amide bond, resulting in an oxyanion bound to a tetrahedral carbon. The terminal amine of the sphingosine moiety is predicted to attract the proton from Asp92, resulting in the collapse of the tetrahedral carbon intermediate, releasing sphingosine and a fatty acid. Finally, the protein catalyst is regenerated.

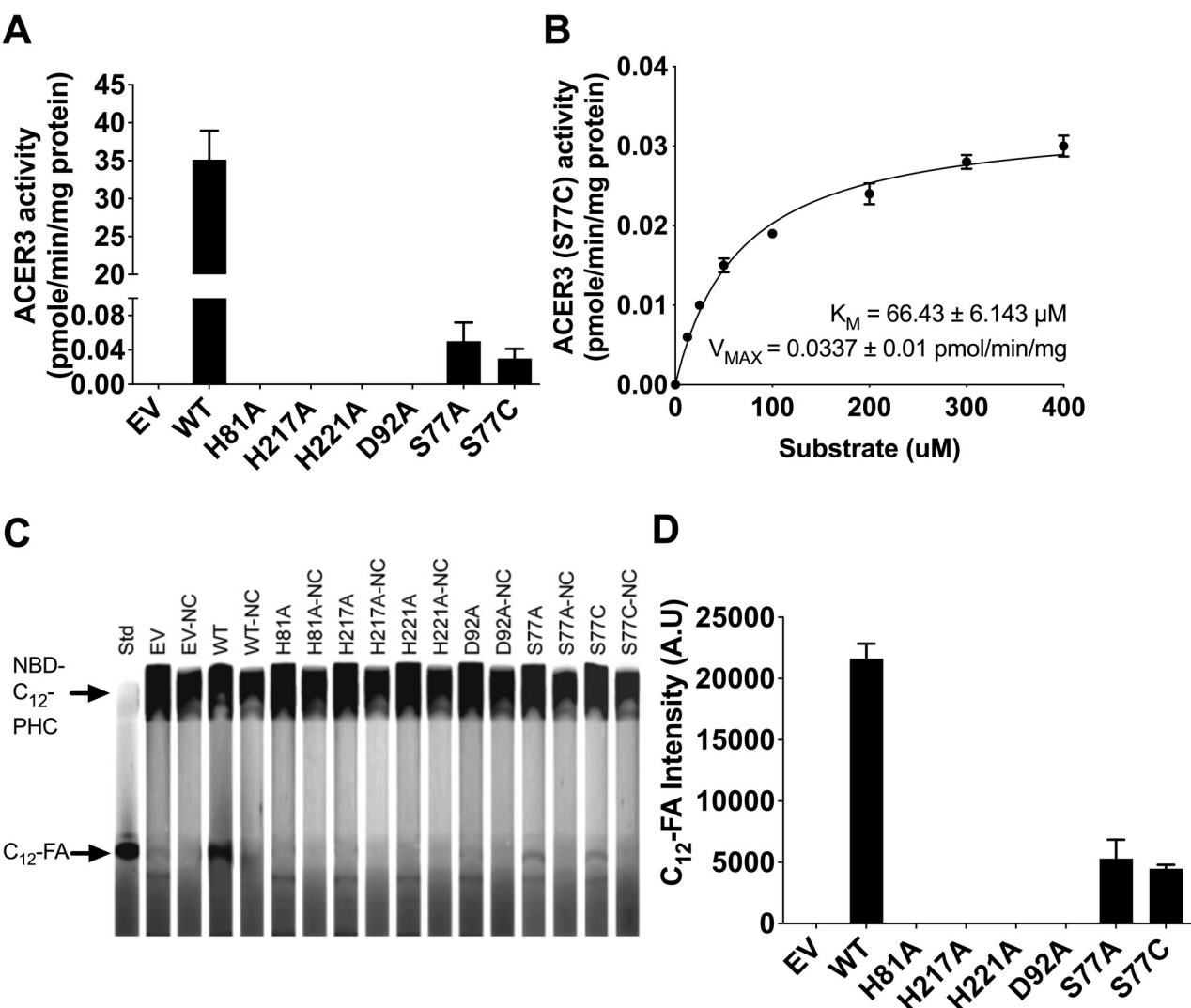

**Fig 3. Role of Ser77 in ACER3 catalytic site.** (A) Microsomes from WT and each mutant ACER3 were subjected to alkaline ceramidase activity using NBD-C$_{12}$-PHC at pH 7.5. The release of the fluorescent product NBD-C$_{12}$-FA from the substrate NBD-C$_{12}$-PHC was detected by HPLC. (B) Microsomes from S77C were measured for in vitro ceramidase activity on various concentrations of NBD-C$_{12}$-PHC at pH 7.5. K$_M$ and V$_{MAX}$ values are indicated. Reactions were conducted with 1 μg of microsome at 37°C for 30 min. Data represent the mean ± S.D. of three independent experiments performed in duplicate. (C) Microsomes from wild-type and each mutant ACER3 were subjected to alkaline ceramidase activity using NBD-C$_{12}$-PHC. The release of the fluorescent product NBD-C$_{12}$-FA from the substrate NBD-C12-PHC was detected by TLC. NC, Negative Control (heated microsomes) (D) Intensity of C$_{12}$-FA in (C) was measured by ImageJ (NIH, 1.53a).

Consistent with this mechanism, the mutations of the three proximal Zn-coordinating residues (His81, His217, and His221) and Asp92 abolished ACER3 activity (Fig 2C–2E). Additionally, substitutions of Ser77 dramatically decreased ACER3 ceramidase activity, suggesting the important role of Ser77. Ser77 is predicted to form a hydrogen bond with the oxygen of carbonyl from the amide bond of ceramide to facilitate to position the amide bond of ceramide for nucleophilic attack by the water molecule. Ser77 is also predicted to stabilize the negatively charged oxyanion of the transition state. The proposed role of Ser77 is supported by the pH dependence of the S77C mutant (Fig 3A). At alkaline pH, the cysteine residue is deprotonated and negatively charged thus it would be unable to hydrogen bond with the oxygen carbonyl and would interact unfavorably with the negatively charged oxyanion. Overall, our proposed

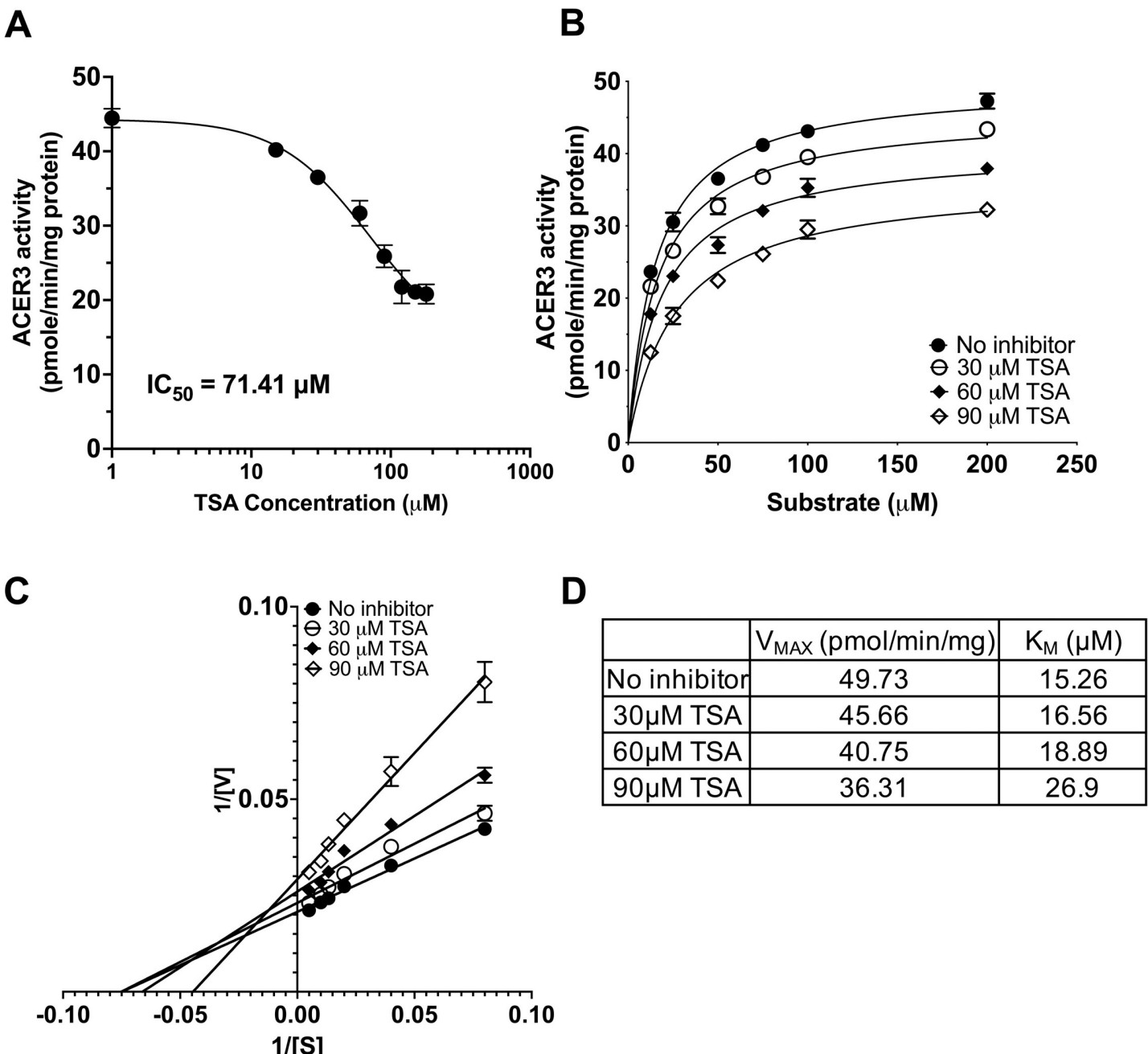

**Fig 4. Inhibitor assay of ACER3.** (A) Microsomes from yeast cells (*ΔYpc1ΔYdc1*) overexpressing wild-type ACER3 were treated with TSA at indicated concentrations before the microsomes were subjected to alkaline ceramidase activity assays using a fixed concentration (75 μM) of NBD-C$_{12}$-PHC. The release of the fluorescent product NBD-C$_{12}$-FA from the substrate NBD-C$_{12}$-PHC was detected by HPLC. The IC$_{50}$ value of TSA was estimated by nonlinear regression using Prism Graphpad software (Prism 9.3.1, San Diego, CA). (B), (C), and (D) Microsomes were measured for *in vitro* ceramidase activity on various concentrations (15 to 180 μM) of NBD-C$_{12}$-PHC in the presence of different concentrations (30 to 90 μM) of TSA. Michaelis-Menten curves (B) and Lineweaver-Burk plots (C) were generated by Prism Graphpad software. The K$_M$ and V$_{MAX}$ values in the absence of any inhibitor or in the presence of different concentrations of TSA were computed (D).

mechanism identifies key roles for all the universally conserved residues of the CREST superfamily in the active site of ACER3.

Importantly, our proposed mechanism does not involve rearrangement of the Zinc-coordinating residues as proposed for ADPRs [13]. In the proposed ADPR mechanism, the

**Fig 5. Suggested ACER3 catalytic mechanism.** The bottom right panel, tetrahedral transition state.

conserved Ser residue is proposed to be involved in ceramide hydrolysis. Specifically, they suggest that the rearrangement of the zinc binding site upon ceramide binding can lead to the direct coordination of Ser77 hydroxyl to the zinc. However, the low but the measurable activity of the S77A and S77C mutants is not consistent with a role for the conserved Ser in metal coordination. Based on the pH sensitivity, it appears it is most likely involved in the stabilization of the transition state oxyanion. In support of this, a similarly positioned Threonine residue in LpxC has been proposed to play a similar role in stabilizing the oxyanion of the transition state [30].

Lastly, as ACERs have emerged as therapeutic targets for cancer [31], this study provides insight into the type of inhibitors that may be suitable for targeting ACERs. Several therapeutic targets, including HDACs, LpxC, peptide deformylase, MMPs, and neutral ceramidase, have similar active sites. Hydroxamate compounds, such as TSA, have proven effective inhibitors for this class of enzymes. Especially, it has been reported that TSA treatment leads to an increase in endogenous ceramides in human cancer cells [32]. In turn, increased ceramides induced cell death, suggesting a potential inhibitory effect on ACERs by TSA in human cells as

well as in test tubes. Given the similar Zn-based active site of ACER3 and the canonical catalytic mechanism we propose, it is likely that hydroxamates, which display strong binding to Zn centers, may be developed as inhibitors of the ACERs for cancer therapy.

## Supporting information

**S1 Fig. HPLC chromatogram of standard NBD-C$_{12}$-FA.** (A) HPLC chromatogram of different concentrations of NBD-C$_{12}$-FA (100 fM, 1 pM, 10 pM, and 100 pM) obtained from HPLC-FLD Fluorescent Detector (Agilent, Santa Clara, CA) set to excitation and emission wavelengths of 467 and 540 nm, respectively. (B) Merged HPLC chromatogram of standard NBD-C$_{12}$-FA (C) Calibration curve for NBD-C$_{12}$-FA. Equation: Y = 1.345*X + 3.189, R$^2$ = 0.9. (TIFF)

**S2 Fig. Comparison between TLC and HPLC used in this study.** Technical comparison of two chromatography used in this study. (TIFF)

**S3 Fig. Linearity of ACER3 activity assay.** (A) Assay linearity with reaction time and three different protein amounts (0.5 μg, 1 μg, and 2 μg). The graph shows NBD-C$_{12}$-FA (pmole) versus the reaction time for each different amount of microsome preparation. (B) Assay linearity with protein amount. The graph shows NBD-C$_{12}$-FA (pmole) versus the amount of microsome for 30 min. Data represent the mean ± S.D. of three independent experiments performed in duplicate. (TIFF)

**S4 Fig. HPLC chromatogram of ACER3 reaction.** Upper panel shows HPLC chromatogram of wild-type ACER3 reaction. The lower panel shows HPLC chromatogram of empty vector reaction. The substrate (NBD-C$_{12}$-PHC) and product (NBD-C$_{12}$-FA) are indicated. Reactions were conducted with 1 μg of microsome at 37˚C for 30 min. (TIFF)

**S5 Fig. The effects of neutral and alkaline ceramidase inhibitors on ACER3 activity.** Microsomes from yeast cells (*ΔYpc1ΔYdc1*) overexpressing wild-type ACER3 were treated with C6-Cer-Urea or DMAPP at indicated concentrations before the microsomes were subjected to alkaline ceramidase activity assays using NBD-C$_{12}$-PHC. The release of the fluorescent product NBD-C$_{12}$-FA from the substrate NBD-C$_{12}$-PHC was detected by HPLC. CTR, non-treated. (TIFF)

## Author Contributions

**Conceptualization:** Lina M. Obeid, Michael V. Airola, Cungui Mao.

**Data curation:** Jae Kyo Yi, Michael V. Airola.

**Formal analysis:** Jae Kyo Yi, Michael V. Airola.

**Funding acquisition:** Lina M. Obeid, Yusuf A. Hannun, Michael V. Airola, Cungui Mao.

**Investigation:** Jae Kyo Yi, Ruijuan Xu.

**Methodology:** Jae Kyo Yi, Ruijuan Xu.

**Project administration:** Yusuf A. Hannun, Michael V. Airola.

**Software:** Michael V. Airola.

**Supervision:** Lina M. Obeid, Yusuf A. Hannun, Michael V. Airola, Cungui Mao.

**Visualization:** Jae Kyo Yi.

**Writing – original draft:** Jae Kyo Yi.

**Writing – review & editing:** Jae Kyo Yi, Michael V. Airola, Cungui Mao.

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
