## [Decision Letter · Decision Letter 0]

31 Mar 2022

PONE-D-22-03848Alkaline ceramidase catalyzes the hydrolysis of ceramides via a catalytic mechanism shared by Zn^2+^-dependent amidasesPLOS ONE

Dear Dr. Mao,

Thank you for submitting your manuscript to PLOS ONE. After careful consideration, we feel that it has merit but does not fully meet PLOS ONE’s publication criteria as it currently stands. Therefore, we invite you to submit a revised version of the manuscript that addresses the points raised during the review process. In your revised manuscript please address as fully as possible the constructive comments of the two reviewers. Please submit your revised manuscript by May 13 2022 11:59PM. If you will need more time than this to complete your revisions, please reply to this message or contact the journal office at plosone@plos.org. Please include the following items when submitting your revised manuscript:A rebuttal letter that responds to each point raised by the academic editor and reviewer(s). You should upload this letter as a separate file labeled 'Response to Reviewers'.A marked-up copy of your manuscript that highlights changes made to the original version. You should upload this as a separate file labeled 'Revised Manuscript with Track Changes'.An unmarked version of your revised paper without tracked changes. You should upload this as a separate file labeled 'Manuscript'.

We look forward to receiving your revised manuscript.

Kind regards,

Israel Silman

Academic Editor

PLOS ONE

Journal Requirements:

"This work was supported, in whole or in part, by National Institutes of Health Grants R01CA163825 (to C.M), P01CA097132 (to Y.A.H and C.M), GM062887 (to L.M.O), and R35GM128666 (to M.V.A.)"

"National Institutes of Health Grants  (to C.M),  R01GM130878 (to C.M and L.M.O), P01CA097132 (to Y.A.H and C.M), and R01GM062887 (to L.M.O)

National Institutes of Health Grants R35GM128666 (to M.V.A.)

URL of National Institutes of Health Grants: https://www.nih.gov/grants-funding

C.M.: decision to publish, preparation of the manuscript and supervision

Y.A.H: supervision and project administration

L.M.O: supervision and project administration

M.V.A.: formal analysis, supervision, data-curation and manuscript preparation"

Reviewers' comments:

Reviewer's Responses to Questions

**Comments to the Author**

1. Is the manuscript technically sound, and do the data support the conclusions?

Reviewer #1: Yes

Reviewer #2: Yes

2. Has the statistical analysis been performed appropriately and rigorously? 

Reviewer #1: Yes

Reviewer #2: Yes

3. Have the authors made all data underlying the findings in their manuscript fully available?

Reviewer #1: No

Reviewer #2: Yes

4. Is the manuscript presented in an intelligible fashion and written in standard English?

Reviewer #1: Yes

Reviewer #2: Yes

5. Review Comments to the Author

Reviewer #1: This is a very nice article in which the mechanism of hydrolysis of ceramides by ACER3 is elucidated. Importantly, a similar mechanism does like operate for other members of the CREST superfamily of integral-membrane hydrolases. The authors use microsomes from yeast mutant cells without endogenous ceramidase activity to show the effect of ACER3 mutants of the enzyme activity. Interestingly, the finding that trichostatin A inhibits ACER3 strongly suggest that medicinal chemistry efforts on trichostatin A structure may render derivatives highly selective for ACER3 over other Zn+2-dependent enzymes.

The article is essentially publishable as it is, but I have an important comment: In Fig S3, NBD-C12-PHC does not seem to be pure, which is important in the context of the article. Why are they so many peaks in the HPLC chromatogram? With such an impure substrate, kinetic analyses afford wrong numbers.

Minor comments.

1. In the Abstract, I would say “Consistent with this mechanism, ACER3 was specifically inhibited by the HDAC inhibitor trichostatin A, a strong zinc chelator.

2. In the methods section, replace extraction buffer by extraction solvent, since a mixture of chloroform/methanol is not a buffer.

3. In the discussion, line 3: “The activated water molecule undergoes a nucleophilic attack on the ceramide amide bond, resulting in an oxyanion bound to a tetrahedral carbon”.

4. Throughout the article, replace hydroxymate by hydroxamate

5. Figure 4. Panel A. It would have been nicer to run a dose response curve and calculate the IC50 for trichostatin A, but, as this would not change the conclusions of the article, it is not strictly necessary to run this experiment. Please do statistical analysis of the TCA bars.

6. Figure 4. Panel B. Two more TCA concentrations (i.e. 30 and >60 muM) would have been nice to see how Km and Vmax change with the inhibitor concentration and further support the type of inhibition. Again, this is not strictly necessary, but it would make a better paper.

7. Fig S1C: More points between 10 and 100 pM C12-FA are necessary to have a solid calibration line.

Reviewer #2: In this study, the author investigated the catalytic mechanism of human alkaline ceramidase 3 (ACER3) by focusing three histidine, one aspartate, and one serine residue, which are suggested to form a metal-dependent active site for lipid hydrolysis and conserved in the CREST superfamily. As a result, Ala mutation of the His and Asp residues completely abolished the activity of ACER3, and mutation of Ser to Ala or Cys showed a significant decrease in activity. The authors suggested that the Ser residue is involved in stabilization of amide bond of ceramide to facilitate the catalytic reaction of ACER3. Furthermore, it was shown that trichostatin A (TSA), an inhibitor of HDAC class I / II, inhibits the activity of ACER3.

Major points

1) In this study, the authors newly established HPLC assay for detection of ceramidase activity. The author described “This newly established method was extremely sensitive and could quantitate NBD-FA levels at far lower levels than conventional TLC-based method. (page 5 line 35-37)”. However, considering that the reader will use this method in the future, a diagram should be presented that specifically shows how sensitive (also quantitative) the detection of ceramidase activity by the HPLC method is compared to TLC.

2) The authors evaluated the activity of ceramidase only by hydrolysis reaction of ceramide. Alkaline ceramidase also catalyzes the reverse hydrolysis reaction, thus, it would be better if the authors examine the effect of point mutations on reverse hydrolysis reaction activity.

3) The author described “This is indicative of an uncompetitive mechanism of inhibition (Fig. 4B and 4C).”; however, Figure 4B requires a Lineweaver–Burk plot or similar plot with multiple concentrations of inhibitor. In addition, IC50 value of TSA is also required.

6. PLOS authors have the option to publish the peer review history of their article (what does this mean?). If published, this will include your full peer review and any attached files.

Reviewer #1: No

Reviewer #2: No

---

## [Author Response · Author response to Decision Letter 0]

28 Jun 2022

Reviewer#1:

1. In the Abstract, I would say “Consistent with this mechanism, ACER3 was specifically inhibited by the HDAC inhibitor trichostatin A, a strong zinc chelator.

- It has been revised as suggested.

2. In the methods section, replace extraction buffer by extraction solvent, since a mixture of chloroform/methanol is not a buffer.

- It has been replaced as suggested.

3. In the discussion, line 3: “The activated water molecule undergoes a nucleophilic attack on the ceramide amide bond, resulting in an oxyanion bound to a tetrahedral carbon”.

- We have revised the sentence.

4. Throughout the article, replace hydroxymate by hydroxamate

- They have been replaced.

5. Figure 4. Panel A. It would have been nicer to run a dose response curve and calculate the IC50 for trichostatin A, but, as this would not change the conclusions of the article, it is not strictly necessary to run this experiment. Please do statistical analysis of the TSA bars. Two more TSA concentrations (i.e. 30 and >60 muM) would have been nice to see how Km and Vmax change with the inhibitor concentration and further support the type of inhibition. Again, this is not strictly necessary, but it would make a better paper.

- We have performed ACER3 activity assays in the presence of different concentrations of TSA and determined the effects of TSA on the Km and Vmax of ACER3 and the IC50 value of TSA.

6. Fig S1C: More points between 10 and 100 pM C12-FA are necessary to have a solid calibration line.

- Two more points (20 pM and 50 pM) have been added and the plot was recalculated.

7. In Fig S3, NBD-C12-PHC does not seem to be pure, which is important in the context of the article. Why are they so many peaks in the HPLC chromatogram? With such an impure substrate, kinetic analyses afford wrong numbers.

- NBD-C12-PHC was pure. The peaks may represent the other products resulting from the action of other enzymes as microsomes not purified ACER3 protein were used for enzymatic reactions. 

Reviewer#2:

1. In this study, the authors newly established HPLC assay for detection of ceramidase activity. The author described “This newly established method was extremely sensitive and could quantitate NBD-FA levels at far lower levels than conventional TLC-based method. (page 5 line 35-37)”. However, considering that the reader will use this method in the future, a diagram should be presented that specifically shows how sensitive (also quantitative) the detection of ceramidase activity by the HPLC method is compared to TLC.

- We agree that we need to be more careful when describing the newly established method. From the supplementary data (Fig. S1 and S2), we confirmed that the HPLC assay can measure fM of C12-FA (1.6 LU). This can’t be achieved by most TLC-based methods including ours. The previous review paper (Beate Fuchs et al, 2011) had nicely introduced the pros and cons of the TLC-based method. Based on the paper, the limitation of fatty acid detection by TLC is around 3 ng-100 ng. However, our setting can detect up to 0.037ng of NBD-C12-FA. We added another supplementary figure to compare the TLC and HPLC-based methods in terms of sensitivity and quantification. (Fig S2)

2. The authors evaluated the activity of ceramidase only by hydrolysis reaction of ceramide. Alkaline ceramidase also catalyzes the reverse hydrolysis reaction, thus, it would be better if the authors examine the effect of point mutations on reverse hydrolysis reaction activity.

- Our unpublished data indicate that distinct from the yeast alkaline ceramidases, mammalian alkaline ceramidases, including ACER3, do not catalyze the reverse reaction of ceramidase. As such, we can not evaluate the effect of point mutations on the reverse reaction by ACER3.

3. The author described “This is indicative of an uncompetitive mechanism of inhibition (Fig. 4B and 4C).”; however, Figure 4B requires a Lineweaver–Burk plot or similar plot with multiple concentrations of inhibitor. In addition, IC50 value of TSA is also required.

- We have performed ACER3 activity assays in the presence of different concentrations of TSA, which allowed us to include the Lineweaver-Burk plot and IC50 value of TSA and determined its inhibition type (mixed inhibition) via the Lineweaver-Burk plot. 

Additional edits:

1. We revised the format of the manuscript according to the instruction you provided.

2. Funding information has been removed from the manuscript.

3. Our funding statement is as follows;

National Institutes of Health Grants:

R01GM130878 (to C.M and L.M.O)

P01CA097132 (to Y.A.H and C.M)

R01GM062887 (to L.M.O)

URL of National Institutes of Health Grants: https://www.nih.gov/grants-funding

C.M.: decision to publish, preparation of the manuscript and supervision

Y.A.H: supervision and project administration

L.M.O: supervision and project administration

4. We confirmed ORCID for all authors.

5. We included captions for all supplementary figures.

6. We have added additional references and checked them as suggested.

---

## [Editor Report · Decision Letter 1]

4 Jul 2022

Alkaline ceramidase catalyzes the hydrolysis of ceramides via a catalytic mechanism shared by Zn^2+^-dependent amidases

PONE-D-22-03848R1

Dear Dr. Mao,

We’re pleased to inform you that your manuscript has been judged scientifically suitable for publication and will be formally accepted for publication once it meets all outstanding technical requirements.

Kind regards,

Israel Silman

Academic Editor

PLOS ONE
---

## [Editor Report · Acceptance letter]

23 Aug 2022

PONE-D-22-03848R1 

Alkaline ceramidase catalyzes the hydrolysis of ceramides via a catalytic mechanism shared by Zn^2+^-dependent amidases 

Dear Dr. Mao:

I'm pleased to inform you that your manuscript has been deemed suitable for publication in PLOS ONE. Congratulations! Your manuscript is now with our production department. 

Kind regards, 

on behalf of

Prof. Israel Silman 

Academic Editor

PLOS ONE